# SIRT1 Activation by *Equisetum arvense* L. (Horsetail) Modulates Insulin Sensitivity in Streptozotocin Induced Diabetic Rats

**DOI:** 10.3390/molecules25112541

**Published:** 2020-05-29

**Authors:** Csaba Hegedűs, Mariana Muresan, Andrea Badale, Mariann Bombicz, Balázs Varga, Anna Szilágyi, Dávid Sinka, Ildikó Bácskay, Mihaela Popoviciu, Ioan Magyar, Mária Magdolna Szarvas, Erzsébet Szőllősi, József Németh, Zoltán Szilvássy, Annamaria Pallag, Rita Kiss

**Affiliations:** 1Department of Pharmacology and Pharmacotherapy, Faculty of Medicine, University of Debrecen, Nagyerdei krt. 98, H-4032 Debrecen, Hungary; csaba.hegedus.1983@gmail.com (C.H.); andreabadale@gmail.com (A.B.); bombicz.mariann@pharm.unideb.hu (M.B.); varga.balazs@pharm.unideb.hu (B.V.); dr.szilagyi.anna@med.unideb.hu (A.S.); nemeth.jozsef@med.unideb.hu (J.N.); szilvassy.zoltan@med.unideb.hu (Z.S.); 2Department of Preclinical Disciplines, Faculty of Medicine and Pharmacy, University of Oradea, 1st December Square 10, 410068 Oradea, Romania; marianamur2002@yahoo.com (M.M.); magyar_nelu@yahoo.com (I.M.); 3Department of Pharmaceutical Technology, Faculty of Pharmacy, University of Debrecen, Nagyerdei krt. 98, H-4032 Debrecen, Hungary; sinka.david@pharm.unideb.hu (D.S.); bacskay.ildiko@pharm.unideb.hu (I.B.); 4Department of Medical Disciplines, Faculty of Medicine and Pharmacy, University of Oradea, 1st December Square 10, 410068 Oradea, Romania; elapopoviciu@yahoo.com; 5Institute of Food Technology, Faculty of Agricultural and Food Sciences and Environmental Management, University of Debrecen, Egyetem tér 1, H-4032 Debrecen, Hungary; sebestyen.magdolna@agr.unideb.hu (M.M.S.); szzsoka83@gmail.com (E.S.); 6Department of Pharmacy, Faculty of Medicine and Pharmacy, University of Oradea, 1st December Square 10, 410068 Oradea, Romania; annamariapallag@gmail.com

**Keywords:** SIRT1, diabetes mellitus, insulin resistance, diabetic cardiomyopathy, *Equisetum arvense* L., streptozotocin

## Abstract

BACKGROUND: *Equisetum arvense* L., commonly known as field horsetail is a perennial fern of which extracts are rich sources of phenolic compounds, flavonoids, and phenolic acids. Activation of SIRT1 that was shown to be involved in well-known signal pathways of diabetic cardiomyopathy has a protective effect against oxidative stress, inflammatory processes, and apoptosis that are the basis of diseases such as obesity, diabetes mellitus, or cardiovascular diseases. The aim of our study was to evaluate the antidiabetic and cardioprotective effects of horsetail extract in streptozotocin induced diabetic rats. METHODS: Diabetes was induced by a single intraperitoneal injection of 45 mg/kg streptozotocin. In the control groups (healthy and diabetic), rats were administered with vehicle, whilst in the treated groups, animals were administered with 50, 100, or 200 mg/kg horsetail extract, respectively, for six weeks. Blood glucose levels, glucose tolerance, and insulin sensitivity were determined, and SIRT1 levels were measured from the cardiac muscle. RESULTS: The horsetail extract showed moderate beneficial changes in blood glucose levels and exhibited a tendency to elevate SIRT1 levels in cardiomyocytes, furthermore a 100 mg/kg dose also improved insulin sensitivity. CONCLUSIONS: Altogether our results suggest that horsetail extract might have potential in ameliorating manifested cardiomyopathy acting on SIRT1.

## 1. Introduction

*Equisetum arvense* L. (*Equisetaceae* family), commonly known as field horsetail is a perennial fern. The photosynthetic, heavily branched sterile stems usually are produced between late spring and late autumn. The sterile stems (*Equiseti herba*) represent the medicinal product of the plant mentioned in European Pharmacopoeia (Ph. Eur.8) [1]. The extracts of this species may be used for internal therapy as infusion, alcoholic extract, capsules, or as ointments for external use. The extracts of *Equisetum arvense* L. are rich sources of phenolic compounds, flavonoids, and phenolic acids, with antioxidant capacity and reducing power. [2,3,4,5]. In addition, most widely known phytochemical compounds, like alkaloids, phytosterols, tannins, and triterpenoids have been isolated [4,6]. Numerous studies have described distinct biological effects of *Equisetum arvense* L. extracts, for instance antibacterial, antifungal, antioxidant [7,8], anti-inflammatory [9], neuro- and cardioprotective [6,10], and antiproliferative properties were attributed to horsetail. [4,8,11]. The modulatory effects of horsetail extract on endothelial cells exposed to hypertonic medium are different and dose dependent. It should be noted, that at high doses, *Equisetum arvense* L. extracts had prooxidant effects, induced apoptosis, and decreased IL-6 secretion [8]

Sirtuins are NAD^+^ dependent histone deacetylases (SIRT1, SIRT3, and SIRT5) or ADP-ribosyl transferases (SIRT4 and SIRT6), which are found in many species from bacteria to humans [12,13]. In mammals, seven isoforms of sirtuins have been identified that differ in substrate specificity and intracellular localization. Accordingly, in the nucleus the SIRT 1, 2, 6, and 7, in the mitochondria SIRT 3, 4, and 5, and in the cytoplasm SIRT1 and 2 [14]. Sirtuins are expressed in several tissues (brain, spinal cord, dorsal root ganglia, hypothalamus, liver, pancreatic ß-cells, skeletal muscles, and adipocytes) and play important functions in a variety of biological processes like cellular stress response, DNA repair, genome stability, cell cycle regulation, and cell survival, cellular homeostasis, particularly in metabolic processes, inflammation, oxidative stress, and aging [15,16,17,18,19,20]. The most studied sirtuin is SIRT1 that is involved in the regulation of a large variety of biological functions: it regulates the glucose and lipid metabolism, improves insulin sensitivity, inhibits cell apoptosis, ameliorates the inflammation, protects the cells against the oxidative stress and has antiaging properties, improves the endothelial and cardiac functions, possesses neuroprotective effects in chronic neurodegenerative diseases, positively influences autophagy, and regulates oncogenic factors [21,22,23]. However, while the antidiabetic effect of SIRT1 has been extensively researched, there is few data concerning the role of SIRT1 in the pathogenesis of diabetic cardiomyopathy (DCM). Diabetic cardiomyopathy, one of the most serious complications of diabetes, has been in the focus of research in recent times [24,25,26,27]. Several hypotheses support that different signal pathways are involved in the morphological and molecular alterations of cardiomyocytes; however, there is a consensus on the role of hyperglycaemia, oxidative stress, fibrosis, and apoptosis in the development of DCM. However, though signal transduction pathways such as ERK1/2/Homer1a/SIRT1, AMPK/SIRT1, SERCA2a/UPR/SIRT1, FOXO3a/SIRT1, NF-κB/SIRT1, eNOS/SIRT1 play crucial role in the pathophysiology of DCM, the exact pathomechanism of the disease has not been elucidated [28]. Literature data therefore suggest that SIRT1 plays a significant role in the development and evolution of DCM, presumptively through its antidiabetic, antioxidant, anti-inflammatory, antiproliferative, and anti-apoptotic effects. Furthermore, SIRT1 activators (STACs) have been in the focus of research for over 20 years. Numerous molecules possessing SIRT1 agonist effect were identified during the last decades. One of the most potent compounds is resveratrol, a natural polyphenol of red wine. Nonetheless, many other natural compounds have proven to act as STACs: butein, piceatannol, isoliquiritigenin, fisetin, paeonol, icariin, apigenin, and quercetin [29,30,31,32]. However, the low solubility and bioavailability of natural STACs have demanded the development of synthetic SIRT1 agonists. More than 14,000 similar molecules have been discovered and synthetized, many of which have been tested in preclinical animal models, whilst others are in different clinical trial phases [29,33]. Nonetheless, the natural compounds extracted from traditional plants continue to be of interest for researchers.

Considering that horsetail plays an important role for instance in the oxidative stress response mechanism and disposes of cardiovascular protective effects in which it resembles SIRT1, one of the most important endogenous regulators of glucose and redox homeostasis, the aim of the present study was to investigate if horsetail treatment is able to exert its cardioprotective effects through SIRT1 activation. 

## 2. Results

### 2.1. Microscopic Examination of Equisetum arvense L.

*Equisetum arvense L.* is difficult to differentiate from other related *Equisetum* species that can produce potentially toxic alkaloids. Unfortunately, the macroscopic identification is very difficult, hence for the precise identification we have chosen the microscopic method. In the studied populations the sterile stems of *E. arvense L* possess 6–18 edges/ridges. In the edges, underneath the silicified chlorenchyma tissue develops assimilated palisade that forms the cortical parenchyma. In the parenchyma vallecular canals that are situated in extensive aeriferous areas, in various forms, like channels, gaps, circles, all arranged orderly. The central cylinder begins with a pericycle, and the cells are small and closely joined together. In the fundamental parenchyma of the marrow there are a multitude of vascular bundles, all arranged in a ring. In the vascular bundle the phloem tissue is positioned just beneath the pericycle and it is more developed than the xylem tissue that surrounds the carinal canal filled with water (Figure 1). The stem’s ramifications have 4 deep edges, with silicified chlorenchyma on the top, and underneath well-developed assimilated palisade parenchyma. Consistent with the literature data the central cylinder presents four vascular bundles, without carinal canal (F.R.X., Ph.Hg. VIII., Eur.Ph. 7th Edition) [1,2,3] (Figure 2).

### 2.2. Ultra-High Performance Liquid Chromatography (UHPLC) Analysis of Horsetail Extract

To determine the phenolic compounds concentration contained in the applied *Equisetum arvense* L. extract UPLC analysis was performed. Results of the analysis of phenolic compounds by HPLC are shown in Table 1. Chromatograms of the *E. arvense* L. extracts were recorded at 350 nm (Figure 3). Predominant flavonoids identified in these analyses were chlorogenic acid (1.735 µg g^−1^ dry mass), ferulic acid (0.355 µg g^−1^ dry mass), quercetin-3-*O*-glucoside (27.13 µg g^−1^ dry mass), and quercetin-3-*O*-rutinoside (36.52 µg g^−1^ dry mass). Our results identified that 1 g *Equisetum arvense* L. dry mass contains 63.65 µg quercetin. Therefore, the quercetin concentration in 1g horsetail extract used in our experimental protocol was 795.625 µg. The applied *Equisetum arvense* L. extract doses quercetin amount was identified as follows: in 50 mg/kg horsetail extract 39.78 µg, in 100 mg/kg 79.56 µg and in 200 mg/kg 159.3 µg. Conclusively, we identified that quercetin is the most important phenolic compound of these horsetail extract that is present in a notable concentration in 1 g horsetail dry mass.

### 2.3. Effect of Horsetail on Body Weight

To evaluate if horsetail treatment can enhance impaired weight gain associated with streptozotocin (STZ) treatment, we measured body weight twice per week throughout the study. At the start of the study, there was no statistically significant difference between body weights of the groups (Figure 4, Table 2). After the STZ treatment, the diabetic groups gradually started to gain less weight compared to the healthy rats. Diabetic Control (DC) animals showed significantly decreased body weight from day 3, horsetail treated animals had a significantly lower weight from day 6 compared to Healthy Control (HC) rats. The 100HT (treatment with 100 mg/kg horsetail extract) and 200HT (treatment with 200 mg/kg horsetail extract) group showed a significant increase compared to the DC animals from day 17 until the end of the experiment. 

### 2.4. Effect of Horsetail on Blood Glucose

To evaluate if horsetail treatment can improve elevated blood glucose associated with STZ treatment, we measured blood glucose daily throughout the study. Blood glucose taken in the morning hours is shown in the form of 5-day averages for better representability and more reliable statistical analysis (Figure 5). In the STZ treated groups blood glucose significantly elevated from one day after the diabetes induction. During the six-week period, blood glucose levels did not show changes within the various treatment groups, but the horsetail treated animals showed a statistically significant improvement compared to the DC rats.

### 2.5. Effect of Horsetail on Glucose Tolerance

To assess the effect of horsetail over the glucose intolerance associated with STZ treatment, oral glucose tolerance test (OGTT) was performed at week 4. In response to the glucose load during the OGTT, area under the glucose curve was significantly elevated in all STZ treated groups (Figure 6B), implying decreased glucose tolerance. Statistically significant improvement was seen only in the 100HT group compared to both the diabetes control, and the other horsetail treated animals. In addition, for a better illustrative presentation the data are presented as time-dependent graph (Figure 6A).

### 2.6. Effect of Horsetail on Insulin Tolerance

To assess the effect of horsetail over the impaired insulin response, insulin tolerance test (ITT) was performed at week 5. In response to the glucose load during the ITT, area under the glucose curve was significantly elevated in all STZ-treated groups (Figure 7B), implying decreased glucose tolerance. The effect of horsetail extract on blood glucose levels measured during ITT are presented in Figure 6A. Statistically significant improvement was seen only in the 100HT group compared to the diabetes control rats. 

### 2.7. Effect of Horsetail on Fasting Plasma Insulin

To appreciate the STZ treatment effect on pancreatic beta cell insulin secreting capacity, moreover to evaluate if horsetail could enhance the impaired insulin production associated with STZ treatment, at the end of the experiment fasting plasma insulin was determined from the blood samples. However, all STZ treated groups had a significantly reduced plasma insulin compared to the healthy controls, as a result of STZ treatment the pancreatic B-cells function was partially preserved (Figure 8).

### 2.8. Effect of Horsetail on Adiposity

To assess whether horsetail had any effect on adipose tissue development, adiposity was determined at the end of the experiment as the sum weight of retroperitoneal and epididymal white adipose tissue (WAT) normalized to body weight (Figure 9). At the end of the experimental period, all diabetic groups lost a statistically significant amount of fat. Horsetail treatment showed no effect in this parameter.

### 2.9. Effect of Horsetail on Heart Weight Index

After sampling of animal tissues, the mass of insulin-sensitive tissues such as heart, liver, and abdominal white fat was measured. We observed that the hearts of STZ-treated animals showed higher values compared to the healthy control, and therefore the heart index that represents the heart weight normalized to the body weight of the animals was calculated, and the results were statistically evaluated. (Figure 10). Statistically significant increase of heart weight index was observed in DC and 200HT groups.

### 2.10. Effect of Horsetail on SIRT1 Levels

The aim of our study was to evaluate the role of sirtuins, particularly SIRT1, in the pathomechanism of diabetes and diabetic cardiomyopathy, and in addition to investigate the effect of *Equisetum arvense* L. extract effects in pathological conditions. Western blot analysis was used to determine SIRT1 protein expression from cardiac tissue, whose results are presented in Figure 11. When standardized to Histone H3 housekeeping protein, a marked decrease was observed in all STZ-treated groups compared to healthy control (HC). According to the statistics, groups of DC and 50HT showed similar trends in decrease (*p* < 0.01), while 100HT group differed only with *p* < 0.05, compared to HC group. In addition, the highest dose of horsetail extract (200HT) produced the weakest SIRT1 protein expression (*p* < 0.001). Therefore, we consider that the 100 mg/kg dose of horsetail extract might activate SIRT1 and intervene with signal pathway that involve SIRT1 activity; however, the treatment period was proven to be too short to produce significant effect compared with the diabetic control. 

## 3. Discussion

Diabetes mellitus is the most widespread metabolic disease in the world and its incidence and prevalence are constantly increasing. Diabetes mellitus is characterized by persistent hyperglycaemia often associated with insulin resistance, which is therefore responsible for the complications of the disease such as neuropathy, artheriopathy, kidney dysfunctions, and cardiomyopathy. The therapy of diabetes mellitus has not been entirely resolved, notwithstanding the introduction of new classes of antidiabetic agents in clinical practice in recent years [34,35,36]. Therefore, extensive research is underway to investigate new possible signaling pathways involved in the pathomechanism of diabetes mellitus that may allow the development of more advanced and effective antidiabetic agents or which might prevent or slow down the progression of diabetes or ameliorate the complications of the disease [37,38,39,40]. Officinal herbs, especially the ones possessing a well-known antidiabetic activity such as fenugreek, cinnamon, sage, berries, or turmeric, attract growing attention as agents in primary, or auxiliary antidiabetic therapeutic agents [41,42,43,44]. Field horsetail (*Equisetum arvense* L.) is one of the most commonly used herbs that have been medically applied since ancient times. Its popularity is unbroken up to this day since it widely occurs all around the globe, and can be used in diverse ways. Field horsetail contains multiple active compounds, and is particularly rich in polyphenols, flavonoids, saponins, dietary fibres, vitamins A, E, and C, potassium, calcium, and silicates. Owing to its wide variety of active chemicals, horsetail is used in numerous fields of medicine. It is applied internally and/or externally to stop nasal, pulmonary, or gastric haemorrhages, but also as a diuretic, and in the treatment of ulcers, rheumatic arthritis, and poorly and/or slowly healing wounds. Ancient and some recent notes from Asia also take note of the antidiabetic activity of horsetail [45,46]. Therefore, we have chosen this plant as the subject of our experiment, to investigate its effectiveness in improving changes of blood glucose, and to what extent it might be able to alleviate the symptoms of prediabetes and diabetes such as insulin resistance. We hypothesized that since field horsetail contains flavonoids and other biologically active ingredients, its beneficial effects might be the result of the increase of SIRT1 expression. Our results showed that the ethanol extract of horsetail has different effects on blood glucose and insulin resistance in various doses (Figure 5, Figure 6 and Figure 7). Fasting plasma insulin levels were low, but present in the STZ-treated rats, which means the model achieved a diabetic condition with insufficient insulin production (Figure 8). At 50 mg/kg, horsetail extract has no effect on hyperglycaemia or insulin resistance, while at 200 mg/kg the expected therapeutic effect was not achieved. However, the 100 mg/kg dose significantly reduced the blood glucose levels and improved the whole-body insulin sensitivity (Figure 5 and Figure 7). There is an established link between SIRT1 and glucose metabolism. Wang et al. reported that reduced SIRT1 can lead to hepatic insulin resistance [47,48], and SIRT1 activation alleviates hepatic insulin resistance in obesity [49]. In adipocytes SIRT1 enhances GLUT4 translocation and consequent glucose uptake [47]. SIRT1 is also present in the pancreatic ß-cells where it can increase the glucose stimulated insulin release [47,50]. Selective overexpression of SIRT1 in ß-cells improves glucose metabolism in mice [51], while selective deletion causes glucose intolerance [52]. Data indicates SIRT1 is also able to protect ß- cells against apoptosis [47,53]. Furthermore, in our study the horsetail extract induced an improved body weight gain compared to the diabetic control, chiefly in the 100 mg/kg and 200 mg/kg dose, but failed to reach the body weight of DC rats by the end of the six weeks long experimental period (Figure 4 and Table 2.). Moreover, our results point that the abdominal white adipose tissue weight was significantly reduced in each STZ-treated groups compared to the healthy control (Figure 9). The lessened abdominal WAT weight might be the result of STZ therapy and the consequent relative insulin deficiency, nevertheless, horsetail extract treatment shows a moderate tendency to overcome this impact that may be attributable to the insulin sensitivity improvement property of horsetail. Diabetic cardiomyopathy, one of the most frequent and severe complications of both type I and type II diabetes mellitus, might develop in the early or later phases of the disease, and its prevention and treatment is not solved up to the present time [24]. The most prominent features of DCM are the left ventricle hypertrophy and the gradually declining diastolic function, which leads finally to decompensated heart failure. In the pathogenesis of DCM in addition to the well-known mechanisms, processes like inflammation, metabolic disorders, or oxidative stress may be involved [25,26]. The molecular basis of diabetic cardiomyopathy is not completely elucidated yet. Several hypotheses explain the pathomechanism of DCM such as endoplasmic reticulum (ER) stress, nitro-oxidative stress, mitochondrial dysfunctions, autophagy, apoptosis, and/or post-translational and post-transcriptional alteration of specific structural and signaling proteins [54,55]. Reactive oxidative species (ROS) and advanced glycation end products (AGEs) generated by oxidative stress response are responsible for disturbances in structural and signaling protein folding and functions, dysfunctions of myocyte calcium regulation, the endoplasmic and sarcoplasmic reticulum damage, followed by the decreased activity of sarcoplasmic/endoplasmic reticulum calcium ATPase (SERCA2a) pump [24,54,56,57]. Furthermore, SIRT1 activity is altered due to oxidative stress. SIRT1 plays important roles in the pathogenesis of DCM, exerting their beneficial effects by interfering with various signal pathways. SIRT1 was shown to upregulate ERK, an anti-apoptotic MAP kinase [58] and deacetylates p53, reducing apoptosis [59]. AGEs play a central role in DCM with impairing key enzymes such as AMPK, SERCA2, and contributing to the formation of ROS and subsequent cell damage. SIRT1 was reported to be able to activate or upregulate these enzymes [24,60]. Research projects are underway to develop more advanced therapeutic strategies for the management of DCM, so it would be possible to achieve satisfactory results without adverse effects and harmful drug interactions. For the same reason there is increasing scientific attention towards the traditional herbs and their bioactive compounds, which can be put in use as food supplements or in the development of new drug molecules [61,62,63,64]. We found that the heart weight index, that is the heart weight normalized to body weight, increased significantly in diabetic controls and in the 200HT group compared to the healthy control rats, while the animals treated with 50 mg/kg and 100 mg/kg horsetail extract showed no difference in heart weight index compared to controls (Figure 10). These results might suggest that the STZ diabetes model developed cardiomegaly, which the 200 mg/kg horsetail extract was not able to prevent. Furthermore, we found that compared to healthy controls, SIRT1 level significantly decreased in diabetic controls, and horsetail extract in none of the applied doses was able to increase it effectively (Figure 11). To the best of our knowledge, our research group was the first to study the SIRT1 activator effect of horsetail extract in STZ-induced diabetic animal model. However, we must point out that the *Equisetum* extract in 100 mg/kg concentration showed a tendency of elevation in SIRT1 level in the cardiac muscle. We are aware that due to the small number of samples statistical analysis might not give us such a solid conclusion as more samples would, but we must note that the clear tendency of SIRT1 elevation was seen in the horsetail extract dose that proved most effective in improving insulin sensitivity. We assume that in this concentration the *Equisetum* extract might improve the symptoms of DCM through SIRT1 that plays an important role in several molecular signaling pathways in cardiac muscle [28]. Determining the targets and mechanism involved in this effect will require further experiments. 

## 4. Materials and Methods

### 4.1. Ethics

*Equisetum arvense* L. plants were used in our study. The plant samples arose from unpolluted areas of the spontaneous flora of Oradea (Oradea, Bihor county, Romania) and were carefully collected and selected in 2018, and then provided by Dr. Annamária Pallag (Department of Pharmacy, University of Oradea, Faculty of Medicine and Pharmacy, 1st December Square 10, Oradea, 410068, Romania). A specimen of the species is kept in the Herbarium of Pharmacy Department Oradea, code: UOP 04012.

The experiment was conducted in accordance with the guiding principles of European Community and the University of Debrecen Ethics Committee for Animal Research for the care and use of experimental animals (ethical code number: 29/2017/DE MÁB, the date of approval of ethical submission: 6 March 2018). 

### 4.2. Animals

Thirty-eight, 6–7-week-old male Wistar rats weighing 175–200 g were used throughout the study (TOXI-COOP ZRT., Budapest, Hungary). The animals were housed at 22–24 °C and in 50%–70% relative humidity. The lighting was set to alternating 12–12 h light/dark periods. The rats received standard laboratory chow (S8106-S011 SM R/M-Z+H; ssniff Spezialdiäten GmbH, Soest, Germany) and tap water ad libitum.

After one-week acclimatization period the animals were randomly divided into five groups. Healthy controls (*n* = 6) received standard chow and tap water (healthy control, HC). The additional four groups (*n* = 8 each) representing the diabetic groups, received the same standard chow and tap water, nevertheless were given an intraperitoneal dose of 45 mg/kg streptozotocin (STZ) to induce diabetes according to the protocol described below. After the STZ treatment, depending on the applied therapeutic dose the animals were divided into the following groups: diabetic control (DC) treated with vehicle, 50HT, 100HT, and 200HT groups. In these groups the animals were given 50, 100, or 200 mg/kg *Equisetum arvense* L. (horsetail) extract, respectively, dissolved and administered in 1 mL tap water, for six weeks via oral gavage once per day. 

### 4.3. Induction of Diabetes Mellitus

Diabetes induction was done with a single intraperitoneal dose of 45 mg/kg streptozotocin (STZ) (Sigma-Aldrich, Budapest, Hungary) freshly dissolved in 0.1 M, pH = 4.5 cold sodium citrate buffer. 

The healthy controls were injected intraperitoneally with sodium citrate buffer alone in equivalent amount. Blood glucose was checked at 2 h intervals after the STZ injection for 12 h, if blood glucose levels were higher than 31 mmol/L, 1 IU insulin was administered by subcutaneous injection, if blood glucose levels were higher than 25 mmol/L, 0.5 IU insulin was given. However, if blood glucose levels were lower than 2.5 mmol/L, 1 mL 40% glucose solution was given via oral gavage. To reduce the risk of severe hypoglycaemia in the first 24 h the animals also received 5% glucose solution. The rats with blood glucose levels higher than 25 mmol/L five days after STZ injection were considered diabetic and kept for further experiments [65,66]. The rats showing blood glucose levels lower than 25 mmol/L were excluded from the study. Accordingly, the number of animals have changed as follows: total number of animals: n = 35; HC: group n = 6; DC group: n = 6; 50HT group: n = 8; 100HT group: n = 8 and 200HT group: n = 7. 

### 4.4. Microscopic Examination of Equisetum arvense L.

From the collected sterile stems and ramifications of *Equisetum arvense* L. samples, originating from the spontaneous flora of Oradea were made microscopical sections. The microscopic analysis of *Equisetum arvense* L. sterile stems was conducted using the OPTIKA B-383PL light microscope (SC Nitech SRL, Bucuresti, Romania), equipped with Proview digital camera and software. Cross sections were made at the level of freshly harvested stems, according to standard methods [1,2]. The main stem has 3 mm in diameter and the lateral ramifications have 1 mm in diameter. The obtained cross sections were analyzed using the objective 10×. After capturing and saving the images with Proview digital camera and software, the cross sections have not been preserved.

### 4.5. Preparation of Equisetum arvense L. (horsetail) Extract

The horsetail (*Equisetum arvense* L.) extract used in this study was prepared according to the following protocol: 300 mL 70% ethanol was added to 50 g horsetail (it equals a 2:1 volume proportion); the mix was left to stand for 24 h in complete darkness, then filtered. The filtrate was boiled at 96 °C by 120 min. The method resulted 3.5–4 g horsetail extract.

### 4.6. Determination of Phenolic Compounds from Equisetum arvense L. Using UPLC-DAD

#### 4.6.1. Chemicals

Acetonitrile and methanol (HiPersolve for HPLC gradient grade Chromanom VWR, Prolabo, Randore, PA, USA). Ultra-pure water was obtained from a Millipore (Bedfore, MA, USA) Milli-Q plus system. Formic acid, reagent grade 98%–100% ACS (ScharlauChemie S. A., VWR, Prolabo, Randore, PA, USA). Quercetin-3-*O*-glucoside was used as pure standards for identification, it was purchased from Sigma Aldrich Co.

#### 4.6.2. Extraction of Phenolic Compounds

Sample extraction was conducted as previously described [67] with minimum modification. The powdered plant material (5 g) was mixed with 100 mL of extraction solvent (methanol:water = 80:20, *V/V*). The mixture was filtered and centrifuged for 5 min at 4500 rpm. The supernatant was evaporated, with Heidolph Hei-VAP Value rotary evaporator (Schwabach, Deutchland, Germany), then dissolved with extraction solvents. The solvent flow rate was 0.45 mL/min. The crude phenolic extract was then isolated by solid phase extraction method using ENVI-18 SPE tubes (Supelclean). The cartridge was activated by washing with 2 mL of methanol, followed by 2 mL of water for conditioning. Then the diluted phenolic extract was loaded in to the cartridge, and impurities were removed by washing with 2 mL 5% methanol. Finally, the crude phenolic acid was eluted from the cartridge using 2 mL of 80% methanol. The purified extract was evaporated then dissolved with 0.5 mL of the extract solvents. The sample was filtered using syringe filter (0.22 µm, PVDF, FilterBio, Nantong, China) and then it was used for analysis.

#### 4.6.3. UPLC Analysis

Ultra-performance liquid chromatography (HITACHI ChromasterUltraRS, HITACHI, Tokyo, Japan) with photodiode array detector (6430), autosampler (6270) interface (6310) and pump (6170), was used for analysis. UV spectra were taken in the 350 nm wavelength. Ten-µl sample was injected and elution was completed in 15 min. Chromatographic condition was conducted: column: Aquity UPLC BEH Shield RP18 1.7 µm; 2.1 × 50 mm (Waters); oven temperature 30 °C. The mobile phase consisted of solvent A (0.1% formic acid in water; *v/v*) and solvent B (100% acetonitrile; the flow rate 0.45 mL/min (the elution gradient used as follows: 99% solvent A (0–1 min, isocratic elution), then for 12 min, lowering solvent A to 0% (linear gradient), from 12.5 to 13.5 min the gradient returned to the initial composition 99% A and then re-equilibrate the column.

### 4.7. Oral Glucose Tolerance Test (OGTT)

OGTT was carried out on week 4 according to the method of Sunhye Lee et al. with minimum modification [68]. Before the experiment, the animals were fasted overnight, then the basal blood glucose levels were determined via a tail clip. Blood glucose concentration was determined by means of glucometer (Accu-Chek, Roche Diagnostics, Budaörs, Hungary). After basal blood glucose measurements, 2 g/kg glucose was administered via oral gavage, then blood glucose was measured at 15, 30, 60, 90, and 120 min. Glucose tolerance was estimated from area under the glucose curve.

### 4.8. Insulin Tolerance Test (ITT)

ITT was carried out on week 5 according to the method optimized and validated in our institute [69]. Before the experiment, the animals were fasted for 3 h, then the basal blood glucose levels were determined via a tail clip. Blood glucose concentration was determined by means of glucometer (Accu-Chek, Roche Diagnostics, Budaörs, Hungary). After basal blood glucose measurements 0.5 U/kg insulin was administered intraperitoneally, then blood glucose was measured at 30, 60, 90, and 120 min. Insulin tolerance was estimated from area under the glucose curve.

### 4.9. Samples

After 6 weeks the animals were killed by cervical dislocation, left ventricular myocardial tissue, epididymal, and retroperitoneal adipose tissue was removed and stored at −80 °C for subsequent measurements. Therefore, adiposity was determined as the sum weight of retroperitoneal and epididymal white adipose tissue (WAT) normalized to body weight. Plasma insulin level was determined by means of radioimmunoassay (RIA) using commercially available insulin RIA kit (RK 400 M, Institute of Isotopes Budapest, Hungary). Both intra- and inter-assay variations were lower than 5%.

### 4.10. Western Blot

Western blot technique was used to analyze and detect SIRT1 (110 KDa molecular weight) protein from the left ventricle tissues of animals. Samples were suspended in a buffer composed of Tris, 25 mM; NaCl, 25 mM; Na-orthovanadate, 1 mM; NaF, 10 mM; Na-pyrophosphate, 10 mM; Okadaic acid, 10 nM; EDTA, 0.5 mM; PMSF, 1 mM; protease inhibitor cocktail; and distilled water (Sigma-Aldrich, St. Louis, MO, USA), using a homogenizer (IKA-WERKE, Staufen, Germany). Total protein concentration was measured with an automated spectrophotometer (FLUOstar Optima, BMG Labtech, Ortenberg, Germany) and bicinchoninic acid (BCA) assay (Sigma-Aldrich, St. Louis, MO, USA). Fifty µg per well amount of total protein containing samples from the nuclear fraction and protein standards were electrophoretically fractionated using 12% SDS-polyacrylamide gels at 25 mA for 120–150 min, then fractionated proteins were transferred onto nitrocellulose membranes. Membranes were blocked in Tris-buffered saline with Tween 20 (TBS-T) and 3% bovine serum albumin (BSA) for 1.5 h. Subsequently each blot was incubated overnight at 4 °C with anti-SIRT1 antibodies (Abcam Plc., Cambridge, UK), diluted 1:1000 in TBS-T, then incubated with horseradish peroxidase-conjugated secondary antibody (Sigma-Aldrich-Merck KGaA, Darmstadt, Germany). Histone H3 (15 KDa) was used as nuclear housekeeping as internal control. Enhanced chemiluminescent substrate (WesternBright™, ECL, Advansta Inc., Menlo Park, CA, USA) was used to identify bands of SIRT1 and Histone H3 proteins. Detection and analysis was made by a C-Digit^®^ blot scanner with Image Studio Digits ver. 5.2. Software (LI-COR Inc., Lincoln, NE, USA). Data was averaged from two independent experiments (n = 4/group).

### 4.11. Statistics

Data were presented as mean ± SEM. Blood glucose and body weight were analyzed with two-way analysis of variance (ANOVA); additional data were analyzed with one-way ANOVA followed by a modified t-test for repeated measures according to Tukey’s method.

## 5. Conclusions

All things considered, we can say that horsetail extract can be recommended as auxiliary therapy due to its beneficial effect on insulin resistance and blood glucose; however, it also might have a potential in preventing diabetic cardiomyopathy and thus reducing morbidity in diabetes. Nevertheless, the exact mechanisms behind the effects of *Equisetum* extract and by what means it exerts its effect in different organs will need further investigation.

## Figures and Tables

**Figure 1 molecules-25-02541-f001:**
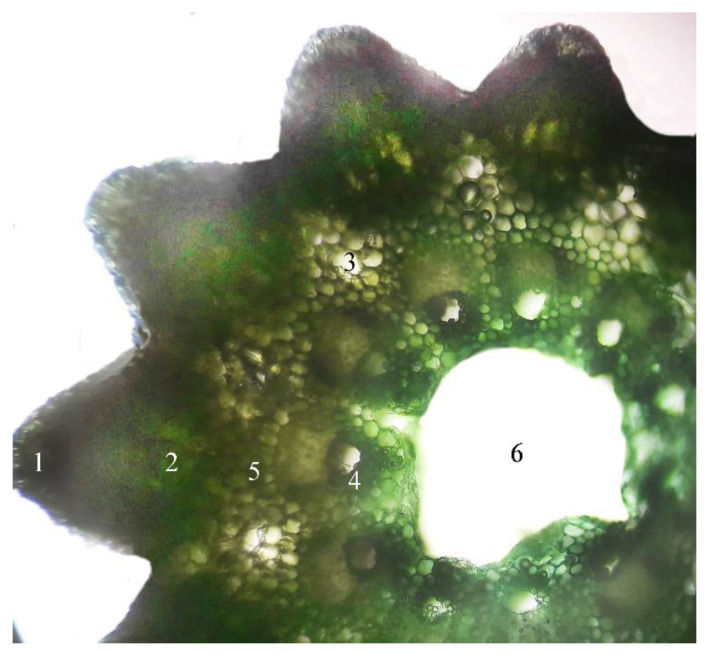
*Equisetum arvense* L. main stem cross section (100×): 1—chlorenchyma; 2—assimilated palisade parenchyma; 3—aeriferous areas in the ground tissue, known as vallecular canals; 4—vascular bundles; 5—cortical parenchyma; 6—pith cavity.

**Figure 2 molecules-25-02541-f002:**
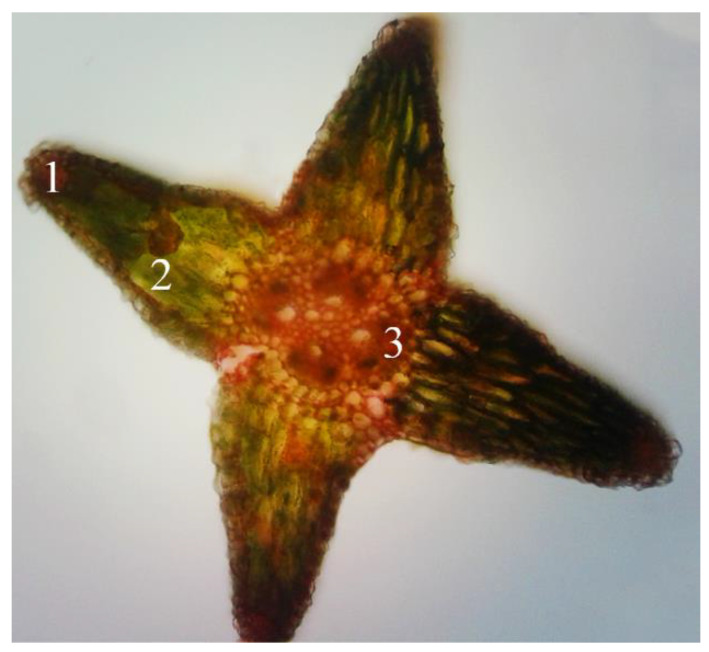
*Equisetum arvense* L. stem ramification cross section (100×). 1—silicified chlorenchyma; 2—palisade parenchyma; 3—leading bundle.

**Figure 3 molecules-25-02541-f003:**
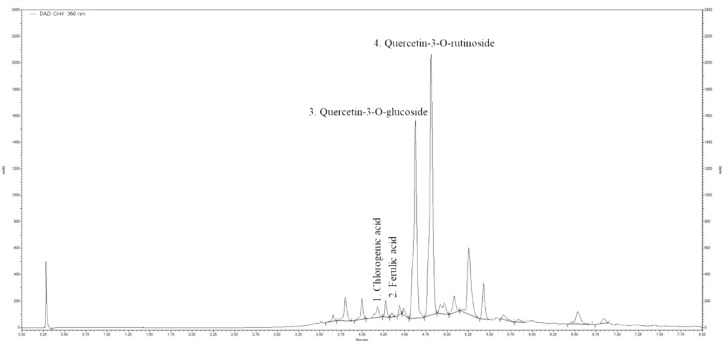
Ultra-High Performance Liquid Chromatography (UHPLC) chromatograms of the extract of horsetail extract at 350 nm. Confirmed by standard.

**Figure 4 molecules-25-02541-f004:**
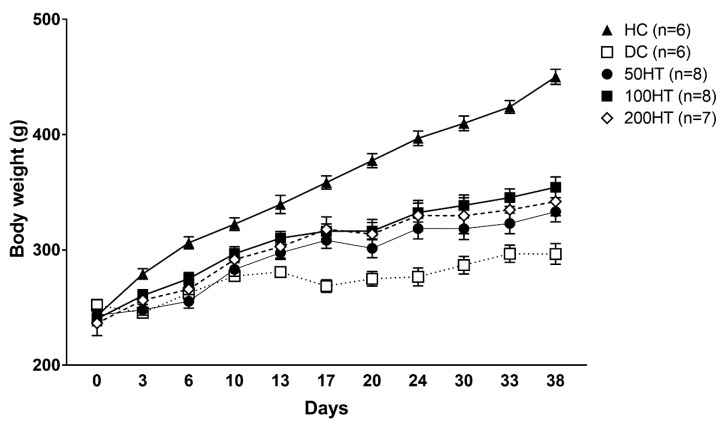
Effect of horsetail on body weight. For clearer representability the symbols showing statistically significant differences are listed in Table 2. In Figure 4 and Table 2 the experimental groups names were abbreviated, as follows: healthy control (HC), diabetic control (DC), 50HT (animals treated with 50 mg/kg horsetail extract), 100HT (animals treated with 50 mg/kg horsetail extract) and 200HT (animals treated with 50 mg/kg horsetail extract).

**Figure 5 molecules-25-02541-f005:**
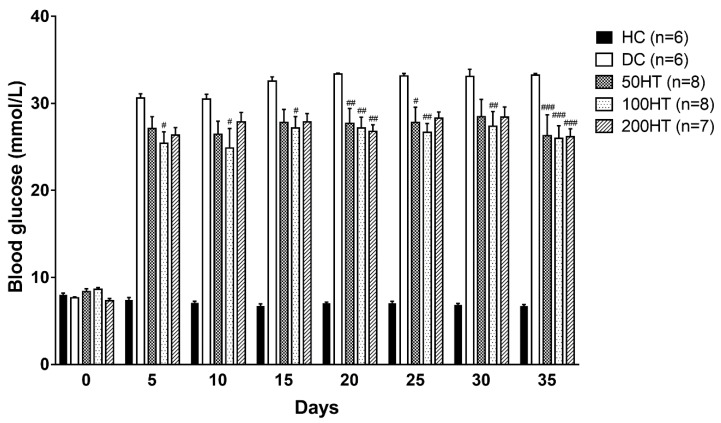
Effect of horsetail on blood glucose. From day 1, all groups showed a significant difference from the HC group (*p* < 0.001). For clearer representability it is not shown with symbols. The #, ##, and ### indicates significant difference from the DC group (*p* < 0.05, *p* < 0.01, and *p* < 0.001, respectively).

**Figure 6 molecules-25-02541-f006:**
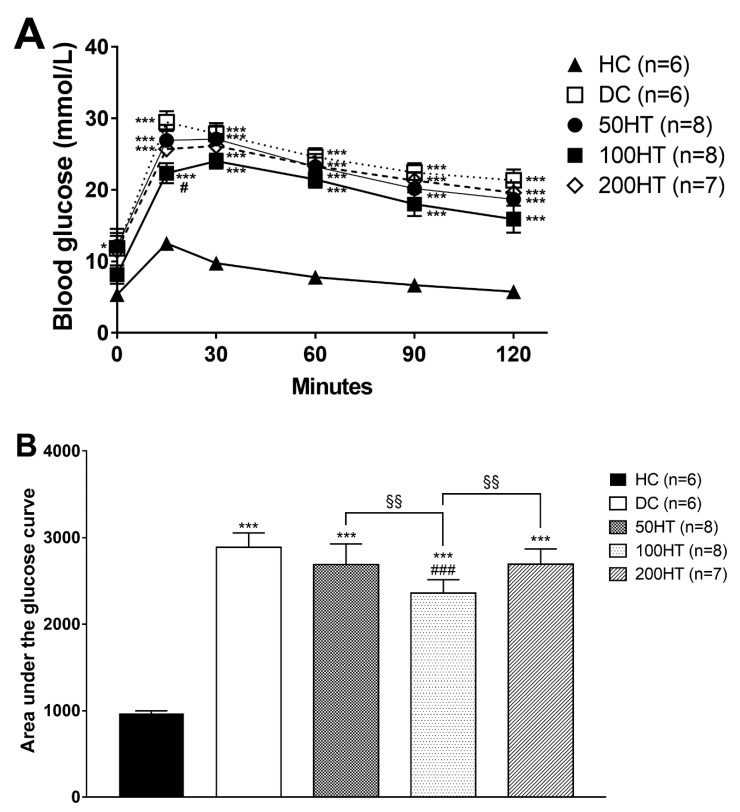
Effect of horsetail on blood glucose levels (**A**) and area under the glucose curve (**B**) during oral glucose tolerance test. The * and *** indicates significant difference from the HC group (*p* < 0.05 and *p* < 0.001, respectively). The # and ### indicates significant difference from the DC group (*p* < 0.05 and *p* < 0.001, respectively). The §§ indicates significant difference between the connected groups (*p* < 0.01).

**Figure 7 molecules-25-02541-f007:**
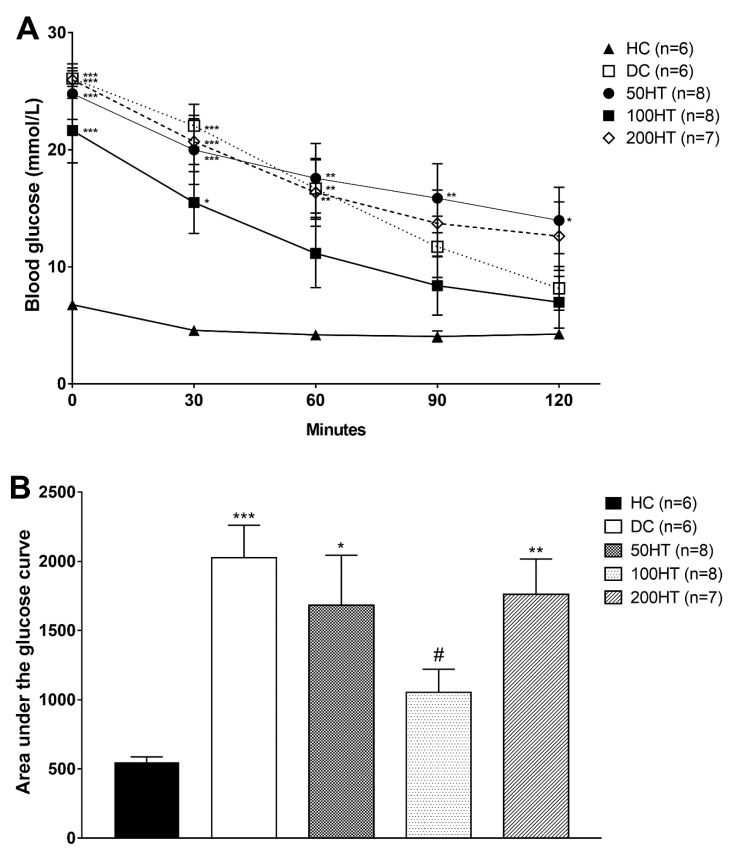
Effect of horsetail on blood glucose levels (**A**) and area under the glucose curve (**B**) during insulin tolerance test. The *, **, and *** indicates significant difference from the HC group (*p* < 0.05, *p* < 0.01, and *p* < 0.001, respectively). The # indicates significant difference from the DC group (*p* < 0.05).

**Figure 8 molecules-25-02541-f008:**
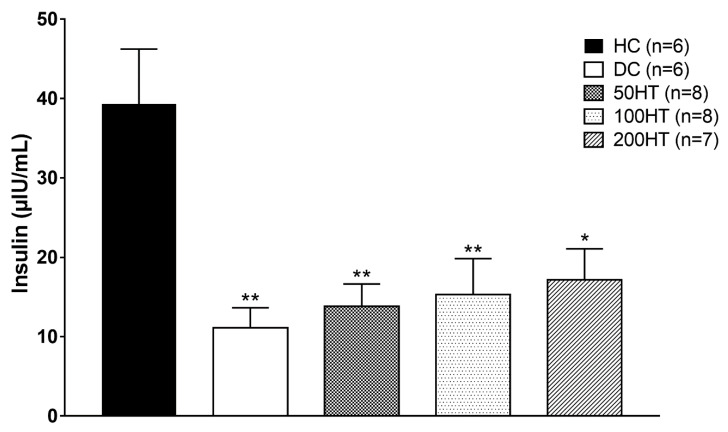
Effect of horsetail on fasting plasma insulin. The * and ** indicates significant difference from HC group (*p* < 0.05 and *p* < 0.01, respectively).

**Figure 9 molecules-25-02541-f009:**
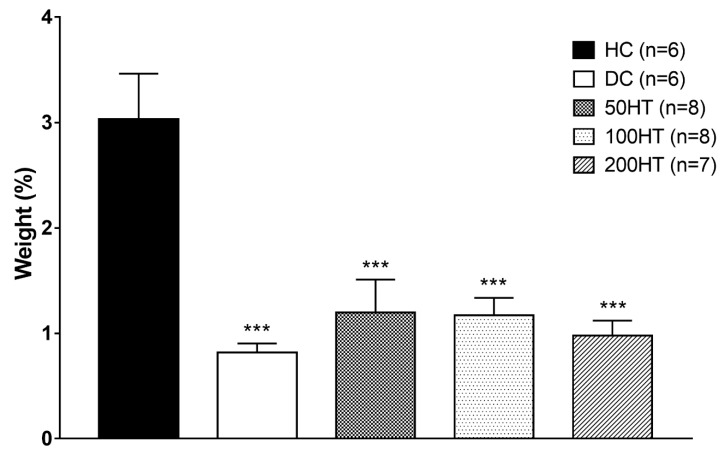
Effect of horsetail on adiposity. The *** indicates significant difference from the HC group (*p* < 0.001).

**Figure 10 molecules-25-02541-f010:**
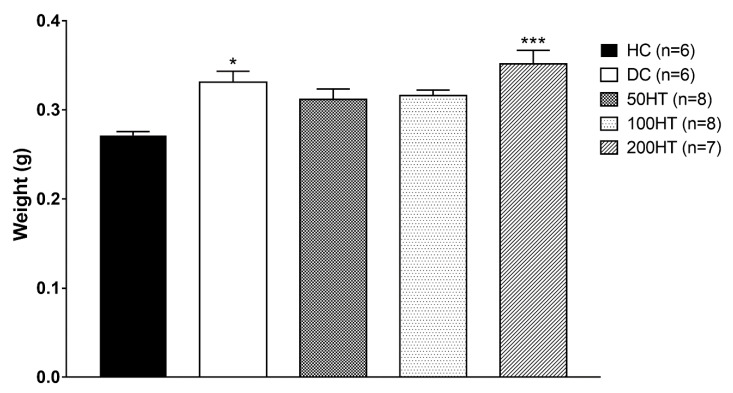
Effect of horsetail on heart weight index. The * and *** indicates significant difference from the HC group (*p* < 0.05 and *p* < 0.001, respectively).

**Figure 11 molecules-25-02541-f011:**
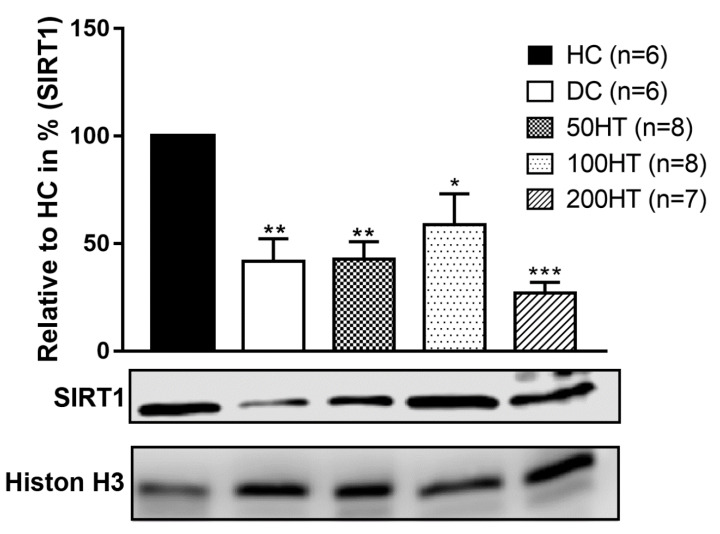
Effect of horsetail on SIRT1 protein expression in left ventricle cardiac tissue. The *, **, and *** indicates significant difference from HC group (*p* < 0.05, *p* < 0.01, and *p* < 0.001, respectively).

**Table 1 molecules-25-02541-t001:** Phenolic compound concentrations in the horsetail extract.

No	RT (min)	Compound	Concentration of Phenolic Compounds (µg g^−1^ Dry Mass)
1	4.18	Chlorogenic acid	1.735
2	4.35	Ferulic acid	0.355
3	4.624	Quercetin-3-*O*-glucoside	27.13
4	4.810	Quercetin-3-*O*-rutinoside	36.52

**Table 2 molecules-25-02541-t002:** Effect of horsetail on body weight. The *, ** and *** indicates significant difference from the HC group (*p* < 0.05, *p* < 0.01, and *p* < 0.001, respectively). The #, ##, and ### indicates significant difference from the DC group (*p* < 0.05, *p* < 0.01, and *p* < 0.001, respectively).

Day	HC	DC		50HT		100HT		200HT	
**0**	242.83 ± 4.46	252.17 ± 4.85		243.29 ± 5.41		240.25 ± 5.87		236.43 ± 10.70	
**3**	279.17 ± 4.51	245.50 ± 3.36	*	248.14 ± 4.70		260.50 ± 4.92		256.43 ± 6.54	
**6**	306.17 ± 5.25	262.50 ± 5.99	**	255.43 ± 5.94	***	275.13 ± 5.28	*	265.86 ± 8.75	**
**10**	322.33 ± 5.58	277.33 ± 4.39	**	283.14 ± 5.99	**	296.75 ± 6.07		291.71 ± 9.29	
**13**	339.50 ± 7.89	280.83 ± 3.26	***	297.57 ± 5.76	**	310.25 ± 5.84		302.86 ± 9.85	*
**17**	358.50 ± 5.77	268.67 ± 5.45	***	308.43 ± 7.03	*** ##	316.25 ± 6.23	** ###	317.86 ± 10.72	** ###
**20**	377.50 ± 6.05	275.00 ± 6.28	***	301.43 ± 8.07	***	316.38 ± 7.41	*** ##	313.71 ± 12.79	*** ##
**24**	396.83 ± 6.35	276.67 ± 7.83	***	318.57 ± 8.99	*** ##	332.38 ± 8.20	*** ###	329.86 ± 13.16	*** ###
**30**	409.83 ± 6.31	286.83 ± 7.60	***	318.43 ± 9.36	***	338.63 ± 9.11	*** ###	329.71 ± 15.59	*** ##
**33**	424.00 ± 5.56	296.67 ± 7.43	***	323.00 ± 8.84	***	345.38 ± 7.56	*** ###	334.86 ± 13.29	*** #
**38**	450.17 ± 6.52	296.50 ± 8.92	***	333.14 ± 8.80	*** #	354.38 ± 9.00	*** ###	342.00 ± 12.33	*** ##

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
