# Peer review of "SIRT1 Activation by Equisetum arvense L. (Horsetail) Modulates Insulin Sensitivity in Streptozotocin Induced Diabetic Rats"

_molecules, 2020, doi:10.3390/molecules25112541_

Round 1
Reviewer 1 Report
This manuscript reported an effect of Equisetum arvense L. (horsetail) on glucose metabolism in STZ-induced diabetic rats. The research could have some scientific impact; however, there are some critical issues which have to be properly addressed at this point.
Major issues:
- In introduction, the description of sirtuins is too broad and the logic is rather unclear why the authors particularly and solely investigated SIRT1 protein levels among many other possible mechanisms of diabetic cardiac myopathy in this manuscript.
- In results, the authors should describe the purpose and sum of major findings of every experiment. It is hard in a current form to follow a scientific logic of this manuscript.
- In results 2.1, the authors suddenly used abbreviations of HC, DC, and HT without explaining what they are. Those should be explained in the legend of figure 4.
- In figures, the number of animals what they actually used in each experiment should be clearly written.
- The authors should show a time-dependent GTT graph along with that of AUC in Figure 6.
- In the DM models with distractive pancreatic b-cells, why did the authors perform insulin tolerance test? A time-dependent ITT graph with that of AUC in Figure 7 should be shown.
- In result 2.7 and discussion, the authors are likely to assume that an increase of heart weight is a definitive sign of diabetic cardiac myopathy in their animal model. No histological or cardiac ultrasound data were provided to prove it. Is there any evidence of their own?
- There is no discussion about the difference between 100 and 200HT treatment. Does 200 HT rather have a toxic effect against DM in rats?
- Lines 216-221 in discussion, it would be difficult to conclude that their DM models are the type 2 DM not type 1 DM model. The authors, moreover, described that they made Type I DM animals in line 302. If it is not essential to distinguish between type 1 and type 2 DM model in this manuscript, all the regarding description should be removed from the text. If they really want to distinguish them, more suitable animal models should have been used in an experimental design.
Minor issues:
- Equisetum arvense L. should be written in italic throughout the manuscript.
- Comma and period are used ambiguously in numbers throughout the manuscript. In scientific international journals written in English, a period should be used at a decimal point.
- There are typos throughout the manuscript.
Author Response
Response to Reviewer 1
We are very grateful to the Reviewer for the mindful review of the manuscript and for the thought-provoking comments that significantly improved the quality of our work. We have extensively revised our manuscript according to the recommendations point-by-point. All changes in the text and the new figures that we have redesigned are highlighted. Please, see the point-by-point answers to your comments below.
Major issues:
Comment 1.: In introduction, the description of sirtuins is too broad and the logic is rather unclear why the authors particularly and solely investigated SIRT1 protein levels among many other possible mechanisms of diabetic cardiac myopathy in this manuscript.
Answer 1.:
Gratefully accepting the observation, the “Introduction” section was extensively revised and the paragraphs referring to SIRT1 were rephrased. Please, find highlighted in the manuscript (Lines: 80-91).
Comment 2.: In results, the authors should describe the purpose and sum of major findings of every experiment. It is hard in a current form to follow a scientific logic of this manuscript.
Answer 2.:
Thank you very much for the suggestion. Authors described in the “Results” section the purpose of each experiment, and the summary of major findings, respectively, as it was requested. Lines: 108-110, 130-131, 139-141, 146-147, 165-166, 177-1178, 181-182, 191-192, 193-194, 203-208, 213-214, 221-225, 231-233, 239-249.
Comment 3.: In results 2.1, the authors suddenly used abbreviations of HC, DC, and HT without explaining what they are. Those should be explained in the legend of figure 4.
Answer 3.:
Thank you for the remark. The paragraph (lines 150-153) and the legend of Figure 4. (lines 157-160) were rephrased, and the explanation of the used abbreviations was included in the sentence to be more accurate.
Comment 4.: In figures, the number of animals what they actually used in each experiment should be clearly written.
Answer 4.:
Thank you for the suggestion. In the legend of each figures the number of animals is included in the revised version. Please, see highlighted in the “Results” section of the manuscript. Lines: 155, 172, 183, 184, 197, 198, 209, 217, 227 and 243.
Comment 5.: The authors should show a time-dependent GTT graph along with that of AUC in Figure 6.
Answer 5.:
Thank you for the commentary. We completed Figure 6 with a time-dependent graph (lines 183 and 185).
Comment 6.: In the DM models with distractive pancreatic b-cells, why did the authors perform insulin tolerance test? A time-dependent ITT graph with that of AUC in Figure 7 should be shown.
Answer 6.:
We definitely agree, and we have to mention that this phenomenon was a huge dilemma regarding the discussion of our manuscript. Despite the fact that the classification of diabetes mellitus has changed in recent years, even now it is based primarily on the insulin-producing capacity of the pancreas and the absolute insulin level in the body. Accordingly, the animal models of DM follow this classification. To induce Type 2 diabetes in animals, the most common method is to feed the animals with high fat diet and supplement the diet with sucrose solution, in this way the animal developing diabetes and insulin resistance. For type 1 DM there are different procedures, all of them based on the almost total destruction of pancreatic beta-cells by chemicals like streptozotocin and alloxan. Relatively new literature data support that administration of low-dose STZ can induces type 2 DM, in which approximately 30% preservation of pancreatic B cells has been observed. In this case authors were curious to investigate the recovery of pancreatic beta-cells. (R. U. Ostrovskaya, et al., 2014; R. U. Ostrovskaya, et al.; 2019, Erdal, N. et al, 2011), nonetheless, other researches prefer the combination of low dose STZ with high fat diet (David André Barrière et al. 2018; Nath S, et. Al 2017; Vatandoust N, Rami F). Other studies have examined the presence of insulin resistance in Type 1 DM. (Bulum T, et al.; Nadeau KJ, et al.; Schauer IE et al.) Consequently, our research team decided to investigate the intensity of insulin resistance in the chosen diabetic animal model. If the Reviewer deems it, the commentary and the mentioned references will be included in the manuscript.
Accepting the suggestion, we supplemented Figure 7 with a time-dependent graph (lines 199 and 201).
Comment 7.: In result 2.7 and discussion, the authors are likely to assume that an increase of heart weight is a definitive sign of diabetic cardiac myopathy in their animal model. No histological or cardiac ultrasound data were provided to prove it. Is there any evidence of their own?
Answer 7.:
We accept the reviewers point, authors agree with the fact that without cardiac ultrasound and/or histological examination of myocardial tissue a cardiac hypertrophy diagnosis cannot be proven, thus the sentence (lines 221-225 and 320-322) was rephrased accordingly. Please, see highlighted in the manuscript.
Comment 8.: There is no discussion about the difference between 100 and 200HT treatment. Does 200 HT rather have a toxic effect against DM in rats?
Answer 8.:
We cannot state with certainty that the 200 mg/kg dose of horsetail extract is toxic. The main crucial parameters of the treated animals were not altered during the experimental period, therefore the toxicity of this dose was not proved, but only the reduced efficacy in the investigated condition. However, our experimental results showed that the effect on heart index and SIRT1 levels was notably worse compared to the other two tested doses. This worsen effect of 200 mg/kg horsetail extract on heart index might be the result of the slower body weight gain of the animals in this experimental group. Regarding to the effects of 200 mg/kg horsetail extract on body weight, blood glucose levels, OGTT, ITT or plasma insulin levels significant differences in comparison with the other used doses were not observed. If the reviewer accepts our answer and forasmuch as request, we could insert this comment in the “Discussion”.
Comment 9.: Lines 216-221 in discussion, it would be difficult to conclude that their DM models are the type 2 DM not type 1 DM model. The authors, moreover, described that they made Type I DM animals in line 302. If it is not essential to distinguish between type 1 and type 2 DM model in this manuscript, all the regarding description should be removed from the text. If they really want to distinguish them, more suitable animal models should have been used in an experimental design.
Answer 9.:
Again, we agree with the reviewer, it is not essential to distinguish between type 1 and type 2 DM, therefore we made the corrections accordingly. The corrections are highlighted in the manuscript at line 274 and 321. Please, also see Comment 6 answer.
Minor issues:
Comment 1.: Equisetum arvense L. should be written in italic throughout the manuscript.
Answer 1.:
Thank you for the remark. The whole manuscript was extensively revised and corrections were made throughout.
Comment 2.: Comma and period are used ambiguously in numbers throughout the manuscript. In scientific international journals written in English, a period should be used at a decimal point.
Answer 2.:
We made the required corrections throughout the manuscript.
Comment 3.: There are typos throughout the manuscript.
Answer 3.:
Thank you for this comment. The whole manuscript was extensively revised and the grammatical and spelling errors were corrected.
Finally, we would like to thank you again for the thoughtful comments. We also believe that based on the comments, the quality of this paper has sig
Reviewer 2 Report
In this paper is introduced the evaluation of the effects of horsetail extracts (Equisetum arvense L.) for 6 weeks in streptozotocin induced diabetic rats. Several metabolic parameters were monitored such as blood glucose, glucose tolerance, insulin sensitivity as well as SIRT1 levels. Global results suggest that horsetail extract may represent a potential treatment, exerting its cardioprotective effects through SIRT1 activation. Experiments look sound, results are well discussed and supporting conclusions; however, minor observations are highlighted for authors´ attention:
ABSTRACT
Along with the document “Equisetum arvense” and “E. arvense” must be written in italics
INTRODUCTION
Lines 65-66
“Accordingly, in the nucleus are located the SIRT 1, 2, 6 and 7; in the mitochondria, SIRT 3, 4 and 5; and in the cytoplasm, SIRT1 and 2 [17].”
Line 76-81
“The SIRT1 activators (STACs) have been in the focus of researches for the last 20 years. Numerous molecules possessing SIRT1 agonist effect were identified during the last decades. One of the most potent compounds is resveratrol, a natural polyphenol of red wine. Nonetheless, many other natural compounds have proven to act as STACs: butein, piceatannol, isoliquiritigenin, fisetin, paeonol, icariin, apigenin and quercetin [29-32]. However, the low solubility and bioavailability of natural STACs have demanded the development of synthetic SIRT1 agonists.”
RESULTS
Line 113 “quercetin-3-O-glucoside and quercetin-3-O-rutinoside.”
Table 1 They should be “Ferulic acid”, “Quercetin-3-O-glucoside” and “ Quercetin-3-O-rutinoside”
Figure 4 Make figures and Tables self-explained (e.g., say what are HC, DC, 50HT, 100HT, etc.)
Line 213 "Therefore, we have chosen this plant as the subject of our ..."
Line 220 Shouldn’t be “a type 1 diabetic condition”?
Line 272 complete “SIRT1”
MATERIALS AND METHODS
Line 294 Please provide the initial age and weight of animals. Also, give the number of animals in each experimental group treated with STZ. Please mention the total duration of the whole experiment, how was the sacrifice performed? and how the extracts were prepared and administered to the rats?
Line 306 Allow proper spacing between words and between values and unitshLine 295, 351, 386 Avoid starting sentences/paragraphs with numbers or abbreviations, e.g., “Thirty eight male Wistar rats …”
Lines 308, 312, 327, 366, 390 use “h” as unit for hours
Line 324 VERY IMPORTANT ...!!
The extract used for the experiment (using EtOH 70%) was not the same as the one analyzed by UPLC (MeOH 80%). Authors should compare how different are both extracts or report only the used one (EtOH 70%). Also, authors should discuss the observed effects and the relation with the phytochemicals present that were administered (There is no discussion at all about this).
Lines 327-328 It is better to provide boiling time “The filtrate was boiled at 96 °C by ?? min.”
Line 336
“Sample extraction was conducted as previously described [79] with minimum modifications.”
Line 340, 345 “… then dissolved with …”
Line 348 What was the solvent flow rate?
Line 388 “… then fractionated proteins were transferred”
Line 397 “… averaged from three independent experiments (n = 4/group).”
Why the group size was changed to four?
REFERENCES
Revise references, they are too many. Some are incomplete (e.g., 5, 14, 41, 64, 70, 74-77), others are older (e.g., 49, 63, etc.) or look reiterative, generic or no significant.
Author Response
Response to Reviewer 2
Firstly, we, the authors of the present manuscript wish to thank you for thoughtful commentary you have provided to improve the quality of the paper. We are very grateful for the time and effort you have devoted to this task. We have extensively revised our manuscript according to the recommendations. All changes in the text and the new figures that we have redesigned are highlighted. Please, see the point-by-point answers to your comments below.
Comment 1.:
ABSTRACT
Along with the document “Equisetum arvense” and “E. arvense” must be written in italics
Answer 1.:
Thank you for the suggestion. The whole manuscript was extensively revised and corrections were made throughout.
Comment 2.:
INTRODUCTION
Lines 65-66
“Accordingly, in the nucleus are located the SIRT 1, 2, 6 and 7; in the mitochondria, SIRT 3, 4 and 5; and in the cytoplasm, SIRT1 and 2 [17].”
Answer 2.:
Thank you for the amendment. The authors corrected the sentence accordingly. Please, see the correction highlighted in the manuscript (line 69).
Comment 3.:
Line 76-81
“The SIRT1 activators (STACs) have been in the focus of researches for the last 20 years. Numerous molecules possessing SIRT1 agonist effect were identified during the last decades. One of the most potent compounds is resveratrol, a natural polyphenol of red wine. Nonetheless, many other natural compounds have proven to act as STACs: butein, piceatannol, isoliquiritigenin, fisetin, paeonol, icariin, apigenin and quercetin [29-32]. However, the low solubility and bioavailability of natural STACs have demanded the development of synthetic SIRT1 agonists.”
Answer 3.:
Thank you very much for the correction. We modified the paragraph accordingly (lines 91-97).
Comment 4.:
RESULTS
Line 113 “quercetin-3-O-glucoside and quercetin-3-O-rutinoside.”
Answer 4.:
Thank you for the remark. Accepting the suggestion, we made the required corrections (lines 134-135).
Comment 5.:
Table 1 They should be “Ferulic acid”, “Quercetin-3-O-glucoside” and “Quercetin-3-O-rutinoside”
Answer 5.:
Thank you very much for the suggestion, the table was modified accordingly (line 143).
Comment 6.:
Figure 4 Make figures and Tables self-explained (e.g., say what are HC, DC, 50HT, 100HT, etc.)
Answer 6.:
Gratefully accepting the observation, the paragraph and the legend of Figure 4. were rephrased, and the explanation of the used abbreviations was included in the paragraph to be more accurate (lines 150-153 and 157-160).
Comment 7.:
Line 213 "Therefore, we have chosen this plant as the subject of our ..."
Answer 7.:
Again, we agree with your suggestion, and correction was made accordingly (line 267).
Comment 8.:
Line 220 Shouldn’t be “a type 1 diabetic condition”?
Answer 8.:
Thank you very much for the comment. According to newer literature data with a single administration of a low dose (45 mg/kg) STZ or repeated administration of an even lower dose (30 mg/kg) of STZ, alone or in combination with a high fat diet, can induce type 2 of diabetes mellitus (R. U. Ostrovskaya, et al., 2014; R. U. Ostrovskaya, et al. 2019; Erdal, N. et al, 2011; David André Barrière et al. 2018; Nath S, et. Al 2017; Vatandoust N, Rami F). Furthermore, other researchers investigate the insulin secreting capacity of the pancreatic beta-cells, or the presence of insulin resistance in the case of the aforementioned animal models. After careful consideration of your and Reviewer 1’s comment, the authors decided that it is not essential to distinguish between type 1 and type 2 DM, therefore we will use in the manuscript simply “diabetes” or “diabetic model”. The corrections were made are highlighted in the manuscript at lines 274 and 321.
Comment 9.:
Line 272 complete “SIRT1”
Answer 9.:
Thank you for the remark. The authors corrected the sentence accordingly (line 329).
Comment 10.:
MATERIALS AND METHODS
Line 294 Please provide the initial age and weight of animals. Also, give the number of animals in each experimental group treated with STZ. Please mention the total duration of the whole experiment, how was the sacrifice performed? and how the extracts were prepared and administered to the rats?
Answer 10.:
Thank you for the thoughtful commentary. Authors fully agree with the reviewer, and the requested additional data were included in the “Materials and Methods” section. Please, find highlighted in the manuscript: lines 347, 373-375, 360, 435 and 359-360.
Comment 11.:
Line 306 Allow proper spacing between words and between values and unitshLine 295, 351, 386 Avoid starting sentences/paragraphs with numbers or abbreviations, e.g., “Thirty eight male Wistar rats …”
Answer 11.:
Again, we totally agree with your suggestion, and corrections were made correspondingly (lines: 364, 347, 413 and 449).
Comment 12.:
Lines 308, 312, 327, 366, 390 use “h” as unit for hours
Answer 12.:
Thank you very much for this remark. In the marked lines the unit of “hour” was modified as specified by the reviewer (lines: 366, 370, 389, 429 and 453).
Comment 13.:
Line 324 VERY IMPORTANT ...!!
The extract used for the experiment (using EtOH 70%) was not the same as the one analyzed by UPLC (MeOH 80%). Authors should compare how different are both extracts or report only the used one (EtOH 70%). Also, authors should discuss the observed effects and the relation with the phytochemicals present that were administered (There is no discussion at all about this).
Answer 13.:
Thank you for the comment. Authors fully agree with the Reviewer; it is a very important issue to be clarified. However, the methanol cannot be used for animal feeding experiment, for the analytical measurement of the active components, extraction with methanol is more advantageous, because in the further sample preparation steps (solid phase extraction) methanol is the solvent. We used 80% methanol because an important principle of the purifying procedure is that the water proportion of the prepared sample should be as low as possible. In view of the solubility of these components (ferulic acid, quercetin-3-O-glucoside and quercetin-3-O-rutinoside) both methanol and ethanol can be applied. There is no difference between them (polarity index of methanol: 5.1 and polarity index of ethanol: 5.2).
Comment 14.:
Lines 327-328 It is better to provide boiling time “The filtrate was boiled at 96 °C by ?? min.”
Answer 14.:
Thank you very much for the suggestion. The authors fully agree with the reviewer, and the boiling time has been written in line 390: “The filtrate was boiled at 96 °C by 120 min.”
Comment 15.:
Line 336
“Sample extraction was conducted as previously described [79] with minimum modifications.”
Answer 15.:
Thank you very much for the suggestion. The authors corrected the paragraph correspondingly (line 398).
Comment 16.:
Line 340, 345 “… then dissolved with …”
Answer 16.:
Thank you for the remark. Corrections in lines 340 and 345 (now 402 and 407) were made accordingly.
Comment 17.:
Line 348 What was the solvent flow rate?
Answer 17.:
Thank you very much for the remark. Authors fully agree, the information was missed, hence we added the solvent flow rate in the “Materials and Methods” section (4.6.2 – line 402).
Comment 18.:
Line 388 “… then fractionated proteins were transferred”
Answer 18.:
Again, the author would express their gratitude for the remark. The suggested modification was made (line 451).
Comment 19.:
Line 397 “… averaged from three independent experiments (n = 4/group).”
Why the group size was changed to four?
Answer 19.:
Thank you very much for the question. In line 397 there was a typing mistake, the average of two independent experiments were used for statistical analysis of Western blot results. The sentence was rephrased as follows: “Data was averaged from two independent experiments (n = 4/group).” In addition, we would like to explain why the group number was changed to four: only 4 animal tissues were randomly selected for molecular biology research, and the average of these two separate analysis was used in the results interpretation. Please, find highlighted in the manuscript (lines 459-460).
Comment 20.:
REFERENCES
Revise references, they are too many. Some are incomplete (e.g., 5, 14, 41, 64, 70, 74-77), others are older (e.g., 49, 63, etc.) or look reiterative, generic or no significant.
Answer 20.:
Thank you for your suggestion. The authors revised the References, and the older and some of incomplete references were removed from the list. Although, Reviewer 3 suggests that the authors mention references to the protocols used (such as OGTT, ITT) and as well as for comparison of the antidiabetic effects of the active ingredients of horsetail extract with other herbs. According to the Reviewer request we totally reedit the References.
Again, we would like to express our gratitude for the hard work you assign for reviewing this manuscript, for your help to rephrased and reedit the manuscript, and providing thoughtful commentary that improved undoubtedly the quality of the paper.
Reviewer 3 Report
The manuscript presented is interesting. There are some suggestions for improving their paper.
Abstract
Should be organized as background, methods, results and conclusions.Their formulation of the abstract is too general.
Methods
Please describe the experimental protocol for control and each study group, otherwise is difficult to follow the results.
The reference for diabetes mellitus induction by streptozotocin is missing. According with who's protocol they did the experiment? Also for the preparation of Equisetum extract, OGTT and ITT. They have to mention the references for these protocols.
Please specify the role of microscopic examination of the plant because the reason for this evaluation is not clear.
They described the method of adiposity evaluation in results. This part should be replaced in Methods chapter.
Discussions
Please see the following references for the turmeric as antidiabetic therapeutic agent:
Molecules. 2019 Feb 27;24(5). pii: E846. doi: 10.3390/molecules24050846.Liposomal curcumin is better than curcumin to alleviate complications in experimental diabetic mellitus. Int J Nanomedicine. 2019 Nov 18;14:8961-8972. doi: 10.2147/IJN.S226790. eCollection 2019.Comparative effect of curcumin versus liposomal curcumin on systemic pro-inflammatory cytokines profile, MCP-1 and RANTES in experimental diabetes mellitus. They should refer to their table and figures when they discuss about their results compared with other studies. All the tables and figures should be included in their dissusions. Conclusions The manuscript has no conclusions. The final conclusios have to be written as a distinct part of the paper, nor as a component of discussions.Author Response
Response to Reviewer 3
We are grateful to the Reviewer for the interest in our work, and for the constructive comments, that will greatly improve our manuscript. We have checked all the comments provided, and have made necessary changes accordingly to the indications. Please find the point-by-point answers below.
Comment 1.:
Abstract
Should be organized as background, methods, results and conclusions. Their formulation of the abstract is too general.
Answer 1.:
Gratefully accepting the observation, we modified the and rephrased the “Abstract” of the manuscript as requested by the Reviewer (lines 31-48).
Comment 2.:
Methods
Please describe the experimental protocol for control and each study group, otherwise is difficult to follow the results.
Answer 2.:
Thank you for the remark. The description of the experimental protocol was extensively revised, and the whole section (4.2.) was rephrased correspondingly (lines 353-360).
Comment 3.:
The reference for diabetes mellitus induction by streptozotocin is missing. According with who's protocol they did the experiment? Also for the preparation of Equisetum extract, OGTT and ITT. They have to mention the references for these protocols.
Answer 3.:
Thank you very much for the comment. The induction of diabetes mellitus was based on protocols of Jianpu Wang et. al. and Sadek KM et al., which were modified to the presented experimental protocol by our research group. Therefore, the references were added to 4.3 subsection (line 363). The OGTT investigation was performed according to method described by Sunhye Lee et al. (lines 425-426), however, the used ITT protocol was developed, optimized and validated by our research team (lines 428-429), as well as the procedure of Equisetum arvense L. extract preparation.
Comment 4.:
Please specify the role of microscopic examination of the plant because the reason for this evaluation is not clear.
Answer 4.:
Thank you so much for the commentary. Equisetum arvense L. is difficult to differentiate from other related Equisetum species that can produce potentially toxic alkaloids. The macroscopic identification is questionable, hence for a more precise and reproducible identification we have chosen the microscopic method. Thenceforth, thanks to the detailing of the microscopic features of the plant, the identification will be much easier. The authors added the purpose of this investigation to the “Results” section (2.1. subsection, lines 108-110). Obviously, if the Reviewer requests, this part will be removed from the manuscript.
Comment 5.:
They described the method of adiposity evaluation in results. This part should be replaced in Methods chapter.
Answer 5.:
Thank you very much for the suggestion. The authors replaced that description to Materials and Methods section. Please, see highlighted in 4.9. subsection (lines 437-438).
Comment 6.:
Discussions
Please see the following references for the turmeric as antidiabetic therapeutic agent:
Molecules. 2019 Feb 27;24(5). pii: E846. doi: 10.3390/molecules24050846.Liposomal curcumin is better than curcumin to alleviate complications in experimental diabetic mellitus. Int J Nanomedicine. 2019 Nov 18;14:8961-8972. doi: 10.2147/IJN.S226790. eCollection 2019.Comparative effect of curcumin versus liposomal curcumin on systemic pro-inflammatory cytokines profile, MCP-1 and RANTES in experimental diabetes mellitus. They should refer to their table and figures when they discuss about their results compared with other studies. All the tables and figures should be included in their dissusions.
Answer 6:
We would like to express our gratitude for the comment and for the suggestions. We have checked the proposed references, that were proved to be particularly useful, therefore we we added the requested articles (References 43 and 44). In addition, according to your suggestion in the Discussion section we referred to our figures and tables. Please, find highlighted in the manuscript: lines 258, 273, 275, 278, 287, 288, 289, 320, 324, 615 and 619.
Comment 7.:
Conclusions The manuscript has no conclusions. The final conclusios have to be written as a distinct part of the paper, nor as a component of discussions.
Answer 7.:
Thank you very much for the commentary. Accepting the suggestion, we supplemented the manuscript with a separate “Conclusions” sections. Please, find it highlighted in the manuscript, lines 465-470.
Finally, we would like to thank you again for the hard and accurate work you provided. We hope that according to your review the quality of our paper has significantly improved.
Round 2
Reviewer 1 Report
The manuscript has improved tremendously in this version; however, there is one more question to be answered.
- In the assessment of adiposity in result 2.8, the author should use the sum weight of retroperitoneal and epididymal white adipose tissue normalized to the body weight (%) to prove that the weight loss is due to significant loss of adiposity, not general emaciation. If the ratio does differ among groups, the author can rightly claim the adiposity change as they described here.
- Comment: Please make sure one more time that the term "Equisetum arvense L." has been all changed to italic.
Author Response
Response to Reviewer 1
We, the authors of the present manuscript wish to thank you for thoughtful commentary you have provided to improve the quality of this paper. We are very grateful for the time and effort you have devoted to this task. We appreciate your constructive remarks and we made corrections accordingly. Please, find the point-by-point answers below.
Comment 1.: In the assessment of adiposity in result 2.8, the author should use the sum weight normalized to the body weight (%) to prove that the weight loss is due to significant loss of adiposity, not general emaciation. If the ratio does differ among groups, the author can rightly claim the adiposity change as they described here.
Answer 1.: Gratefully accepting the observation, we normalized the weight of retroperitoneal and epididymal white adipose tissue to body weight, and after that statistical analysis was performed. The current results are similar to those previously calculated. Therefore, we have inserted the new figure (instead of the previous one) in the manuscript and obviously we have rephrased the corresponding paragraphs in the “Results” and “Material and methods” section, respectively. Please, find highlighted with green at lines: 215, 219 and 439.
Comment 2.: Please make sure one more time that the term "Equisetum arvense L." has been all changed to italic.
Answer 2.: Thank you for the remark. We have extensively revised again the whole manuscript and corrections were made throughout.
Finally, we would like to express our gratitude once more for your constructive comments, and we hope, improved form of the manuscript will be appropriate for publishing.
Reviewer 2 Report
Authors have looked over their manuscript as per the reviewers´ suggestion. However, there are still minor points to be addressed. After this, revised version may be accepted for publication.
It was pointed out that horsetail´s scientific name should be written in italics, i.e. “Equisetum arvense” and “E. arvense”, but NOT “L.”. Also, check missing changes, lines 107, 108 (2X), 132, 377, 386, 391
Check journal rules for figures (i.e., should have title and captions bellow) and tables (i.e., have title and description at the top)
Author Response
Response to Reviewer 2
We are very grateful to the Reviewer for the mindful review of the manuscript and for the thoughtful comments that significantly improved the quality of our work. We are very beholden for the time and effort you have devoted to this task. We appreciate your constructive remarks and we made corrections accordingly.
Comment 1.: It was pointed out that horsetail´s scientific name should be written in italics, i.e. “Equisetum arvense” and “E. arvense”, but NOT “L.”. Also, check missing changes, lines 107, 108 (2X), 132, 377, 386, 391
Answer 1.: Thank you very much for this comment, authors fully agree. Unfortunately, we have missed to made the corrections in the aforementioned lines. We have extensively revised once more the whole manuscript and corrections were made throughout. Please, find highlighted with green at lines 107, 108, 132, 378, 387 and 392.
Comment 2.: Check journal rules for figures (i.e., should have title and captions bellow) and tables (i.e., have title and description at the top)
Answer 2.: Thank you for the remark. Again, we totally agree with the Reviewer, and the corrections were made accordingly. Please, find highlighted with green at lines 144 and 161.
Finally, we would like to express our gratitude once more for the helpful and constructive comments. We hope that according to your review the quality of our paper has significantly improved and may be more suitable for publishing.